# Modelling individual experts in neonatal seizure detection algorithm development

Sergio Morant Gálvez[1], Mark O'Sullivan[1], Brian Henry Walsh[2,3], Alison O'Shea[1]*

[1]NeuroBell Ltd. Core House, Ballincollig, Cork, Ireland.
[2]INFANT Research Centre, University College Cork, Cork, Ireland.
[3]Department of Paediatrics and Child Health, University College Cork, Cork, Ireland.

*Abstract*— Developing algorithms to detect seizures in neonatal electroencephalogram (EEG) signals is an important area of research. Identifying neonatal seizures is a time-consuming process that requires specially trained experts. Most neonatal seizure detection algorithms use supervised learning and require large datasets of labelled EEG for training. However, EEG is a complex physiological signal, and expert annotators often have disagreements when identifying seizures in infants. Most studies with multiple expert annotators compress the annotations down to one 'ground truth' set of labels during algorithm training, this may lead to a loss of valuable information. This study investigates if preserving the disagreement of multiple expert annotators during training improves model performance. Three variations of a deep learning architecture are compared experimentally; each one varies in how annotator disagreements are accounted for. The results indicate that there is value in modelling expert annotations separately in supervised learning algorithms. This study proposes architectures that harness expert variability by learning from both the agreement and disagreement in an open-source dataset of neonatal EEGs.

*Clinical Relevance*— This work demonstrates how a more holistic approach to neonatal seizure detection algorithm development, incorporating opinions of all annotators, improves algorithm results and better reflects the standards of clinical care.

## I. INTRODUCTION

Neonatal seizures are often the first indication of a serious underlying medical condition [1]. Seizures have a reported incidence rate of 1-5 per 1000 neonates, but identifying seizures is challenging as they usually present no clear clinical signs [2]. The gold-standard for the reliable detection of neonatal seizures is continuous multi-channel EEG monitoring, but this requires expertise, time, and costly equipment for set-up, interpretation, and diagnosis. These challenges have prompted the development of automated algorithms for neonatal seizure detection [3], [4], [5].

Currently the state-of-the-art algorithms for neonatal seizure detection use deep learning to automatically extract hierarchical layers of patterns from labelled datasets of EEG. An algorithm that utilizes labelled training data is a supervised

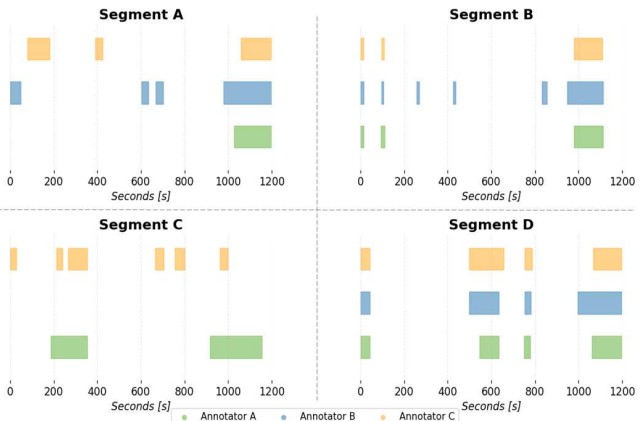

Figure 1 Examples of expert seizure annotations for four EEG segments showing: A) annotators disagreeing on seizure events, B) agreement on some events but one annotator noted additional short seizures, C) two of the three annotators noted seizures, D) a case where there is high agreement among experts.

learning algorithm, and in particular for deep learning models, these algorithms are known to be data hungry [6]. When data and specifically labelled data are limited, innovative development has come from the efficient use of the available data; a recent study utilized pseudo-labelling of test data which increased the amount of labelled data for training and led to improved performance, other studies use data augmentation to artificially generate new training samples [7], [8].

In the domain of neonatal seizure detection, labelled training data comes from expert EEG annotators who review hours of multi-channel EEG and assign binary labels {seizure or no-seizure} across the temporal dimension. Even among experts there can be variations in these binary labels due to the subjective nature of this visual task [9]. Previous works have studied methods of combining these individual expert annotations into a single ground truth label that can be used for training and testing supervised algorithms [10]. Using consensus labels from experts reduces label noise, but a downside of this is that potentially valuable information may

* corresponding author: alison@neurobell.com

be lost. This study seeks to understand the value of learning from individual expert insights and to investigate if modelling expert variability during training can lead to improved model performance [11].

## II. METHODOLOGY

### A. Dataset description

A publicly available dataset of multi-channel neonatal EEG with seizures annotated by three clinical experts is used in all experiments [12]. The dataset contains 112 hours of EEG from 79 infants and was recorded in the NICU of Helsinki University Hospital between 2010 and 2014. The EEG data are represented using a referential montage of 19 electrodes. In this work a bipolar montage of 8 channels is used: [F4-C4; C4-O2; F3-C3; C3-O1; T4-C4; C4-Cz; Cz-C3; C3-T3].

Three clinical experts, each with over 10 years' experience, reviewed the entire dataset and annotated seizure events. A Fleiss' kappa agreement value of 0.767 was reported, representing a very high agreement across three annotators [12]. Despite this high-level of agreement and the expertise of these reviewers, there are still many seizure events, and even entire recordings, where there is disagreement among annotators. In Figure 1, four example EEG segments were selected and the corresponding expert annotations are shown.

During preprocessing the EEG is downsampled from 256Hz to 50Hz, this reduces the size of the data and results in less computational complexity and shorter run time during model development. A high and low pass filter of 0.5Hz and 15Hz respectively are used to filter the EEG signal to a range that includes the dominant frequencies of EEG seizures [13].

The multi-channel EEG is segmented into 16 second windows with a 4 second shift between windows.

### B. Architecture

Three different experimental architectures are proposed, all utilizing the labels from three annotators, but each one employing varying methods for amalgamating the information from individual experts. Seizures are clinically defined as having a duration of greater than 10 seconds [14], but in this study 16 second windows are used as input to the model. To fully represent short seizures and event edges in the training data a sequence-to-sequence architecture was employed. Windows that are partially labelled as seizure were included during training, by representing the label as a temporal sequence of 16 binary values. The base model convolutional neural network (CNN) is shown in Figure 2, and the three variations are:

**A) One model - consensus annotations**
A single CNN model is trained using the consensus seizure annotations from all three expert annotators. Windows of EEG where annotators disagree are not included in the training data.

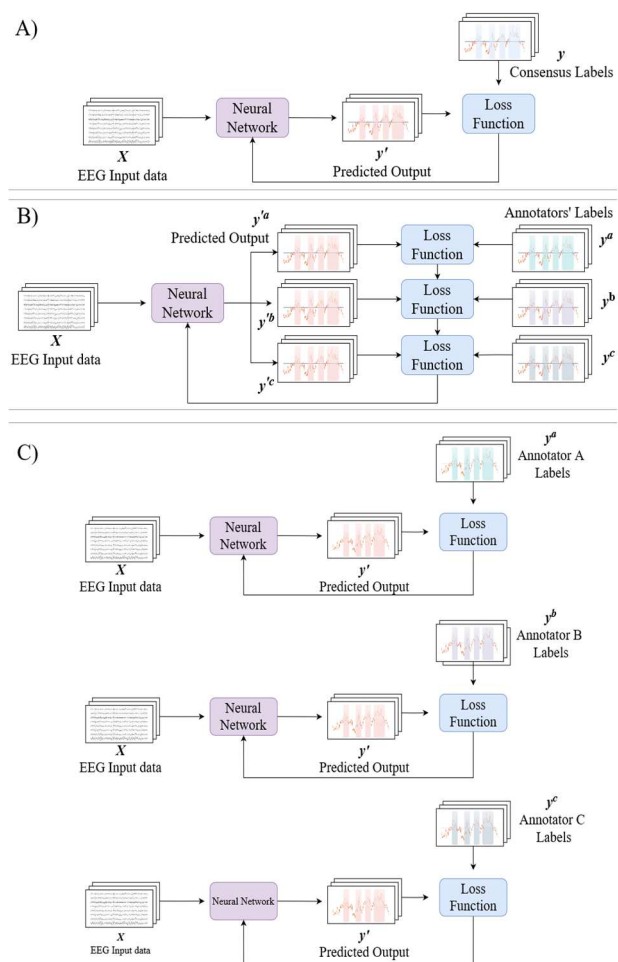

Figure 3 Diagram of the three different approaches: A) one model trained on consensus annotations, B) one model trained to predict the annotation of 3 experts separately, C) three models trained each with a set of labels from a particular expert.

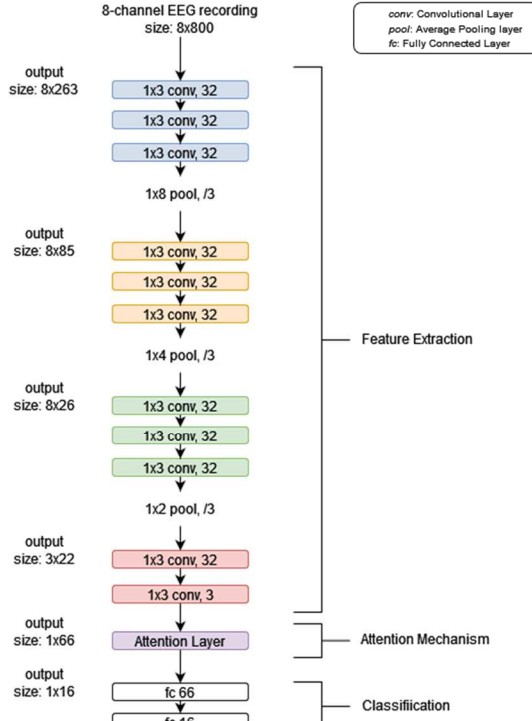

Figure 2 Base model architecture consisting of convolutional blocks followed by an attention layer. The final two fully connected layers map the 16s window of multi-channel EEG input to a 16s vector of per-second seizure decisions using a Sigmoid operation.

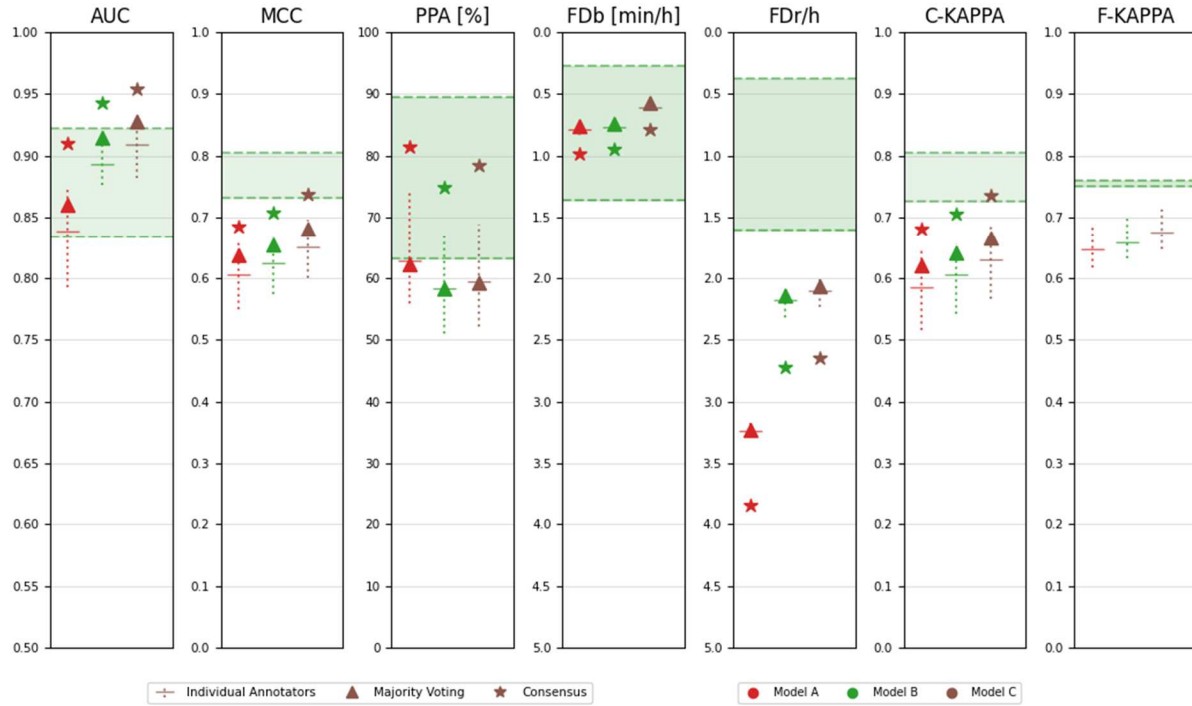

*Figure 4 Results of the LOO experiments for models A, B, C. The green shaded areas show the inter-rater performance of the 3 expert annotators for each metric.*

**B) One model - three heads**

A single CNN where the output layer is adapted to predict multiple labels, one for each annotator, across the 16 second window. In this instance there are 3 annotators, so each "head" predicts the label of that expert for all samples.

**C) Three models - individual annotations**

Three CNNs where each individual model is trained with annotations from one specific expert.

Figure 3 demonstrates how the information from individual expert annotators contributes to the models' weights through the loss function. The degree to which the expertise of individual experts is blended decreases as we move from model A to model C, where the information from experts becomes more separated and distinct.

### C. Experimental Procedure

A leave-one-patient-out (LOO) cross-validation routine was used to test model performance; 79 experiments were run, each used 78 patients for model training and 1 patient for model testing. In each experiment, the test patient was varied until all patients had been tested. All experiments were run using Python with PyTorch on AWS Sagemaker.

In each LOO experiment, 10% of the training patients were used as a validation set. The model was trained for a maximum of 50 epochs using EarlyStopping monitoring AUC on the validation set. A Sequence-to-Sequence approach was used, the model input is an 8-channel 16 second window of 50Hz EEG (8x800 samples) and the model output is a 16 second 1Hz output (16 samples). TverskyLoss [15] was used as a loss function in combination with Adam optimizer and a learning rate scheduler applying a 25% reduction every 5 epochs.

Each EEG window is 16 seconds long with a 12 second overlap between windows. At the model output, there is more than one predicted value for each second of EEG. The average of these values is calculated for each second and a 7-second moving average filter is applied followed by a binary decision threshold at 0.5. For models B and C, the output label is seizure if all three heads or three models predict seizure, otherwise the output label is non-seizure.

### D. Evaluation Metrics

Models are tested by comparing them to the expert seizure annotations. Models are evaluated based on epoch-level performance: AUC (Area Under the Curve), MCC (Matthews Correlation Coefficient), and event-level performance: PPA (Positive Percentage Agreement), FDr/h (False Detection rate per hour), FDb (False Detection burden). Models are also tested by measuring their agreement with the human annotators to calculate an average Cohen's kappa and Fleiss' kappa statistic for each model.

### III. RESULTS

The experimental results are presented in Figure 4. For each model, the results are reported when comparing against:

- the individual experts (the dotted vertical line represents the range across the three annotators, and the solid horizontal line represents the average)

- the majority voting of the three experts (the triangle)

- the consensus annotations, only segments where all three experts agree on seizures are reported as seizures, all other segments are no seizure (the star)

Across almost all metrics, the performance improves when incorporating the information from each annotator separately. The only metric where the model trained with consensus annotations (model A) outperforms the others is PPA, but this

comes at the expense of having a relatively high number of false positives. Figure 4 also shows the models' performances when compared with the agreement between expert annotators, green shading.

## IV. DISCUSSION

This study shows the potential value in modelling individual experts separately and that valuable information is lost when expert annotations are merged. Model architectures that learn from variations between expert annotations report better performance and more closely mimic clinical experts. This improvement is evident in both epoch-level and event-level metrics, indicating that the improvements may translate from a research setting to a clinical environment.

Previous works have indicated that EEG segments with ambiguous labels should be removed from training datasets [10], but in an environment where data are limited and expert annotations are valuable, discarding data should be avoided. The proposed architecture shows that deep learning algorithms can learn from experts, even when there is disagreement. A takeaway from this work is the potential value in diversity and the varying insights of different experts. In Stevenson *et al.*, it is noted that the clinical annotators and patients in the dataset all come from the same clinical center [12], this is a limitation of this work as changes in national guidelines, clinical training, equipment utilized, and many other factors could influence the EEG and expertise across centers. Future work would benefit from assessing the generalizability of these findings by testing on a held-out dataset recorded at a different center.

When the results are compared with the inter-rater agreement between expert annotators, it shows that Model B and Model C approach expert level performance; previously a mean Fleiss' kappa for an algorithm developed on this dataset of 0.646 was reported [3]. Comparing with the expert annotators in this manner, provides a clearer understanding of the potential performance of algorithms with respect to that of clinical experts.

## V. CONCLUSION

This study demonstrates that there is valuable information in the individual labels from expert annotators and that deep learning models can be constructed to harness this information. By training separate networks, each focused on replicating an individual expert, the deep learning task more closely models how human knowledge is represented. The results suggest that future work in this area could benefit from modelling individual experts, rather than compressing their annotations using majority voting or consensus calculations during training. Performance results are limited by the size of the publicly available dataset utilized in this work.

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
