# IEEE COPYRIGHT AND CONSENT FORM

To ensure uniformity of treatment among all contributors, other forms may not be substituted for this form, nor may any wording of the form be changed. This form is intended for original material submitted to the IEEE and must accompany any such material in order to be published by the IEEE. Please read the form carefully and keep a copy for your files.

**Modelling individual experts in neonatal seizure detection algorithm development**

**Sergio Morant Gálvez, Mark O'Sullivan, Brian Henry Walsh, Alison O'Shea**

**2024 IEEE EMBS International Conference on Biomedical and Health Informatics (BHI)**

## COPYRIGHT TRANSFER

The undersigned hereby assigns to The Institute of Electrical and Electronics Engineers, Incorporated (the "IEEE") all rights under copyright that may exist in and to: (a) the Work, including any revised or expanded derivative works submitted to the IEEE by the undersigned based on the Work; and (b) any associated written or multimedia components or other enhancements accompanying the Work.

## GENERAL TERMS

1. The undersigned represents that he/she has the power and authority to make and execute this form.
2. The undersigned agrees to indemnify and hold harmless the IEEE from any damage or expense that may arise in the event of a breach of any of the warranties set forth above.
3. The undersigned agrees that publication with IEEE is subject to the policies and procedures of the IEEE PSPB Operations Manual.
4. In the event the above work is not accepted and published by the IEEE or is withdrawn by the author(s) before acceptance by the IEEE, the foregoing copyright transfer shall be null and void. In this case, IEEE will retain a copy of the manuscript for internal administrative/record-keeping purposes.
5. For jointly authored Works, all joint authors should sign, or one of the authors should sign as authorized agent for the others.
6. The author hereby warrants that the Work and Presentation (collectively, the "Materials") are original and that he/she is the author of the Materials. To the extent the Materials incorporate text passages, figures, data or other material from the works of others, the author has obtained any necessary permissions. Where necessary, the author has obtained all third party permissions and consents to grant the license above and has provided copies of such permissions and consents to IEEE

**You have indicated that you DO wish to have video/audio recordings made of your conference presentation under terms and conditions set forth in "Consent and Release."**

## CONSENT AND RELEASE

1. ln the event the author makes a presentation based upon the Work at a conference hosted or sponsored in whole or in part by the IEEE, the author, in consideration for his/her participation in the conference, hereby grants the IEEE the unlimited, worldwide, irrevocable

permission to use, distribute, publish, license, exhibit, record, digitize, broadcast, reproduce and archive, in any format or medium, whether now known or hereafter developed: (a) his/her presentation and comments at the conference; (b) any written materials or multimedia files used in connection with his/her presentation; and (c) any recorded interviews of him/her (collectively, the "Presentation"). The permission granted includes the transcription and reproduction of the Presentation for inclusion in products sold or distributed by IEEE and live or recorded broadcast of the Presentation during or after the conference.

2. In connection with the permission granted in Section 1, the author hereby grants IEEE the unlimited, worldwide, irrevocable right to use his/her name, picture, likeness, voice and biographical information as part of the advertisement, distribution and sale of products incorporating the Work or Presentation, and releases IEEE from any claim based on right of privacy or publicity.

BY TYPING IN YOUR FULL NAME BELOW AND CLICKING THE SUBMIT BUTTON, YOU CERTIFY THAT SUCH ACTION CONSTITUTES YOUR ELECTRONIC SIGNATURE TO THIS FORM IN ACCORDANCE WITH UNITED STATES LAW, WHICH AUTHORIZES ELECTRONIC SIGNATURE BY AUTHENTICATED REQUEST FROM A USER OVER THE INTERNET AS A VALID SUBSTITUTE FOR A WRITTEN SIGNATURE.

Sergio Morant Gálvez                          02-10-2024

Signature                                     Date (dd-mm-yyyy)

# Information for Authors

## AUTHOR RESPONSIBILITIES

The IEEE distributes its technical publications throughout the world and wants to ensure that the material submitted to its publications is properly available to the readership of those publications. Authors must ensure that their Work meets the requirements as stated in section 8.2.1 of the IEEE PSPB Operations Manual, including provisions covering originality, authorship, author responsibilities and author misconduct. More information on IEEE's publishing policies may be found at http://www.ieee.org/publications_standards/publications/rights/authorrightsresponsibilities.html Authors are advised especially of IEEE PSPB Operations Manual section 8.2.1.B12: "It is the responsibility of the authors, not the IEEE, to determine whether disclosure of their material requires the prior consent of other parties and, if so, to obtain it." Authors are also advised of IEEE PSPB Operations Manual section 8.1.1B: "Statements and opinions given in work published by the IEEE are the expression of the authors."

## RETAINED RIGHTS/TERMS AND CONDITIONS

- Authors/employers retain all proprietary rights in any process, procedure, or article of manufacture described in the Work.
- Authors/employers may reproduce or authorize others to reproduce the Work, material extracted verbatim from the Work, or derivative works for the author's personal use or for company use, provided that the source and the IEEE copyright notice are indicated, the copies are not used in any way that implies IEEE endorsement of a product or service of any employer, and the copies themselves are not offered for sale.
- Although authors are permitted to re-use all or portions of the Work in other works, this does not include granting third-party requests for reprinting, republishing, or other types of re-use.The IEEE Intellectual Property Rights office must handle all such third-party requests.
- Authors whose work was performed under a grant from a government funding agency are free to fulfill any deposit mandates from

that funding agency.

## AUTHOR ONLINE USE

- **Personal Servers**. Authors and/or their employers shall have the right to post the accepted version of IEEE-copyrighted articles on their own personal servers or the servers of their institutions or employers without permission from IEEE, provided that the posted version includes a prominently displayed IEEE copyright notice and, when published, a full citation to the original IEEE publication, including a link to the article abstract in IEEE Xplore. Authors shall not post the final, published versions of their papers.
- **Classroom or Internal Training Use.** An author is expressly permitted to post any portion of the accepted version of his/her own IEEE-copyrighted articles on the author's personal web site or the servers of the author's institution or company in connection with the author's teaching, training, or work responsibilities, provided that the appropriate copyright, credit, and reuse notices appear prominently with the posted material. Examples of permitted uses are lecture materials, course packs, e-reserves, conference presentations, or in-house training courses.
- **Electronic Preprints.** Before submitting an article to an IEEE publication, authors frequently post their manuscripts to their own web site, their employer's site, or to another server that invites constructive comment from colleagues. Upon submission of an article to IEEE, an author is required to transfer copyright in the article to IEEE, and the author must update any previously posted version of the article with a prominently displayed IEEE copyright notice. Upon publication of an article by the IEEE, the author must replace any previously posted electronic versions of the article with either (1) the full citation to the IEEE work with a Digital Object Identifier (DOI) or link to the article abstract in IEEE Xplore, or (2) the accepted version only (not the IEEE-published version), including the IEEE copyright notice and full citation, with a link to the final, published article in IEEE Xplore.

**Questions about the submission of the form or manuscript must be sent to the publication's editor.**
**Please direct all questions about IEEE copyright policy to:**
**IEEE Intellectual Property Rights Office, copyrights@ieee.org, +1-732-562-3966**