# OpenReview forum: "Modelling individual experts in neonatal seizure detection algorithm development"
_IEEE.org/EMBS/BHI/2024/Conference — IEEE BHI'24_

### Official Review · Reviewer_wWYJ · 2024-07-26
**Paper 39 revision. Good approach and description with some weaknesses**

**Overall Rating:** 6
**Confidence:** 3

**Other Quality Metrics:**

(a) Clarity of writing: good
(b) Clinical Significance: weak
(c) Methodological Novelty: good
(d) Experiments and Results: good

**Questions For The Authors:**

No further questions

**Strengths:**

The study employs a comprehensive set of evaluation metrics, including epoch-level (AUC, MCC) and event-level (PPA, FDr/h, FDb) measures, providing a holistic assessment of model performance in a clinically relevant context.
The study avoids discarding potentially valuable data by incorporating all expert annotations, even those with disagreements, maximizing the utility of the limited labelled data available in this domain.
The study treats inter-expert variability as valuable information rather than noise. The authors capture a richer representation of expert knowledge by training separate models to mimic individual experts or predict their annotations.

**Summary Of The Paper:**

Overall, the study presents a compelling case for incorporating expert variability in deep learning models for neonatal seizure detection. The proposed architectures demonstrate promising performance improvements and highlight the potential of this approach to enhance the accuracy and reliability of automated seizure detection systems.

**Weaknesses:**

The study relies on a single publicly available dataset. While the authors employ a leave-one-patient-out cross-validation strategy, the limited size and potential biases within this dataset may affect the generalizability of the findings to other clinical settings and populations.
Computational Complexity: Training multiple models (as in Model C) increases computational complexity compared to a single model approach. This could pose challenges for resource-constrained environments.
While the performance improvements are promising, the study doesn't delve into why modeling individual experts leads to better results. Further investigation into the specific patterns learned by each model could provide valuable insights.
The study focuses primarily on technical evaluation metrics. Further exploration of the clinical implications of the findings, such as how these models might impact clinical decision-making or patient outcomes, would strengthen the study's impact.

---

> ### Author Rebuttal · Authors · 2024-08-30
>
> We want to thank the reviewer for their feedback on our work and we hope to address some of the identified weaknesses in an updated version of the paper, should it be accepted.
>
> We agree that not reporting results on a held-out unseen dataset is a limitation of the testing, this relates to the limitation of having only one dataset publicly available. We believe that using a LOO cross validation routine is the best way to measure performance when limited to a single dataset of patients. We believe that the following addition to the Discussion section would help to clarify this limitation of the work.
> “Future work would benefit from assessing the generalizability of these findings by testing on a held-out dataset recorded at a different center.”
>
> We agree with the reviewer that more work should be done to review the clinical reasoning for improvements seen in these experiments. We hope that future work could include a review of the changes in the types of seizures that were identified in better performing models to see if these correlate with specific kinds of morphologies or etiologies.

---

### Official Review · Reviewer_PyEb · 2024-08-01
**The paper includes promising approach to neonatal seizure detection, however, it requires methodological clarifications and enhanced results interpretation before acceptance**

**Overall Rating:** 6
**Confidence:** 4

**Other Quality Metrics:**

• Clarity of writing. Good

• Clinical Significance. Fair

• Methodological Novelty. Good

• Experiments and Results. Good

**Questions For The Authors:**

Major comments:
- Provide a detailed rationale for the choices made in the preprocessing procedure. Specifically, what considerations influenced your decisions. A clear explanation of the preprocessing rationale will help in understanding the robustness and reproducibility of your methodology, potentially increasing the credibility of your results.
- Similarly, what is the rationale behind changing the window sizes during the architectural adjustments? How do these changes affect the model’s performance and clinical meaningfulness?
- Chapter II in general must be revised: “The consensour output from the 3 models is used at inference” Does this mean only the instances with consensus among the three models are considered? Similarly, clarify what you mean by “by taking the consensus of the 3 labels at inference”. LOO experiments are not clear, rationale of settings should be explained.

Figure comments:
- Figure 1 resolution is scarce. It is recommendable to replace with a higher resolution figure to enhance clarity and readability.
- Figure 2 and Figure 3 should be reordered to ensure they appear sequentially and logically within the text.

Minor comments:
- Briefly introduce the methodology used in your work within the abstract. Including a summary of the methodology in the abstract will provide a clearer overview of your study.
- Avoid using acronyms in the abstract or introduce them if necessary: For example, even though EEG is well known, it should be introduced for clarity.

**Strengths:**

The use of convolutional neural networks (CNNs) in diverse configurations, combined with the inclusion of annotation discrepancies, significantly enhances the training data volume and diversity. This approach leads to a more robust and comprehensive model.

**Summary Of The Paper:**

The paper introduces an innovative method for detecting neonatal seizures. Unlike existing algorithms that rely solely on consensus annotations, this study investigates alternative model configurations that incorporate discrepancies among expert annotations. The paper introduces three different variations of convolutional neural networks (CNNs) and demonstrates that incorporating these discrepancies leads to improved model performance.

**Weaknesses:**

No significant weaknesses are identified relative to the paper’s stated ambitions. However, as the authors themselves suggest, future research should focus on increasing the diversity of annotators. This would ensure that the models are trained on a more varied dataset, potentially enhancing their robustness and applicability across different scenarios.

Moreover, external validation is crucial for the generalizability of the paper’s outcomes. Currently, the model’s performance is assessed based on the available dataset and annotations from the limited group of experts. Conducting external validation would involve testing the model on independent datasets from different institutions or geographical regions, which were not used in the initial training and testing phases. This process would help to confirm that the model's efficacy is not limited to a specific set of conditions or annotators. It would also provide insights into how well the model generalizes to new, unseen data, ensuring its reliability and applicability in a broader clinical context.

---

> ### Author Rebuttal · Authors · 2024-08-30
>
> Many thanks for your review of our manuscript, we appreciate your detailed feedback and your proposed improvements for our work.
>
> Major comments:
> 1.	You suggested that we add a clearer rationale for the preprocessing procedure used in experiments. The preprocessing procedure allows us to filter out frequencies that are outside the range of interest for neonatal seizures and to remove signal artifacts e.g. electrical noise. Additionally, downsampling the EEG reduces the complexity of the input data while maintaining the frequency bands of interest and enough resolution to identify key features in the EEG. Further clarification on these motivations and the purpose of preprocessing steps has been added to the text:
> “During preprocessing the EEG is downsampled from 256Hz to 50Hz, this reduces the size of the data and results in less computational complexity and shorter run time during CNN model development. A high and low pass filter of 0.5Hz and 15Hz respectively are used to filter the EEG signal, the filtered signals includes the dominant range of frequencies of EEG seizures [13].”
>
> [13]	M. Kitayama et al., “Wavelet analysis for neonatal electroencephalographic seizures,” Pediatric Neurology, vol. 29, no.4, pp. 326–333, 2003.
>
> 2.	Apologies if it is not clear in our manuscript but we do not change the EEG window sizes between experiments. We have tried to keep as many hyperparameters as possible stable between the three experiments and the size and dimensionality of the input window sizes is 8 EEG channels by 800samples throughout; 800 samples is 16seconds at 50Hz.
>
> 3.	We have taken the time to revise the text in sections of the Methodology, specifically when it comes to describing the model architecture. Thank you for this comment and we believe this will improve clarity in updated versions of this manuscript.
>
> Figure comments:
> 1.	We have redesigned Figure 1, it represents the same data but we believe that it is represented in a clearer and more readable manner. In updated versions of the manuscript, it will be replaced.
>
> 2.	We have reordered figures 2 and 3 in the text, many thanks for this suggestion.
>
> Minor comments:
> We have updated the abstract to include some information about the experimental methodology and to explain acronyms the first time they are used. We thank the reviewer for these suggestions, they have improved the abstract and the updated version will be included in future versions of this manuscript.

---

### Official Review · Reviewer_CuZH · 2024-08-10
**Modelling individual experts in neonatal seizure detection algorithm development**

**Overall Rating:** 7
**Confidence:** 5

**Other Quality Metrics:**

Clarity of Writing: Good
Clinical Significance: Great
Methodological Novelty: Great
Experiments and Results: Excellent

**Questions For The Authors:**

I noticed that the dataset used comes from a single clinical center. This led me to wonder whether the findings could be generalized to other hospitals with different practices. It would be helpful to understand if there were considerations or limitations around this, and how the model might perform on data from other sources.

 I’m wondering if there were plans or considerations for external validation using independent datasets. Understanding how the model might generalize beyond the current dataset would be valuable.

**Strengths:**

The study focuses on a critical real-world problem—neonatal seizure detection. By demonstrating that incorporating expert disagreement can improve model performance, the research has direct implications for clinical practice, potentially leading to better outcomes in neonatal care.
The paper provides a thorough evaluation of the proposed models using a leave-one-patient-out cross-validation method. This approach ensures that the models are tested rigorously and their performance is reliably assessed across different patients.

**Summary Of The Paper:**

The paper focuses on developing and enhancing algorithms for detecting neonatal seizures using EEG data. Neonatal seizures are critical indicators of serious underlying health conditions in newborns, yet they are challenging to identify due to the subtlety of their clinical signs. EEG monitoring is the gold standard for seizure detection, but it requires specialized expertise, time, and expensive equipment. Automated algorithms have been developed to assist in this process, with the latest approaches leveraging deep learning models trained on labeled EEG data.

However, the labeling process itself is complex and subjective, often leading to disagreements among expert annotators. Traditionally, these discrepancies are resolved by creating a consensus or majority-voted label set, which may lead to the loss of valuable information. This study investigates whether preserving and modeling the individual annotations from multiple experts can improve the performance of seizure detection algorithms.

The paper proposes three different architectures for integrating expert annotations into the training process:

1. Consensus Model: A single CNN trained on the consensus annotations from all experts.
2. Multi-Head Model: A single CNN with multiple output heads, each predicting the labels for one expert.
3. Individual Expert Models: Separate CNNs for each expert, with a consensus taken at inference.

The study utilizes a publicly available dataset of neonatal EEG recordings, annotated by three clinical experts. The experimental results suggest that the models trained on individual expert annotations outperform the consensus-based model in most metrics, indicating that incorporating expert variability into training offers significant benefits.

The study concludes that modeling individual expert opinions can improve the algorithm's performance, more closely mimic clinical decision-making, and ultimately enhance the reliability of automated neonatal seizure detection systems. The findings encourage future work to consider the value of individual expert annotations rather than compressing them into a single ground truth label.

**Weaknesses:**

The dataset used in the study comes from a single clinical center (Helsinki University Hospital). This may limit the generalizability of the findings, as the model’s performance could vary when applied to data from different centers with potentially different clinical practices or patient populations.

The study primarily focuses on internal validation using a leave-one-patient-out cross-validation method. While this is thorough, the lack of external validation on a completely independent dataset limits the ability to assess the true generalizability and robustness of the models.

While the paper presents novel architectures, it does not thoroughly compare these with other state-of-the-art seizure detection algorithms. A more extensive comparison could help establish the relative advantages and limitations of the proposed methods.

---

> ### Author Rebuttal · Authors · 2024-08-30
>
> Many thanks for your review and your feedback on our manuscript. Based on your advice and questions we have included some additional discussion on the limitations and intended future work in our paper.
>
> We agree that utilizing data from a single clinical center is a limitation of this study, but we like many researchers are reliant on publicly available datasets. The dataset utilized in this study is a publicly available dataset of neonatal EEG with the associated seizure annotations also available. Unfortunately, we were not aware of any other publicly available datasets with similar population profiles and seizure annotations when this work was completed. By using only a publicly available dataset, it ensures that others can also replicate these experiments with their own architectures and algorithms. We agree with the reviewers, that having a more diverse dataset, ideally from multiple centers, would be preferable and we propose to add the following to the Discussion to express this limitation of the work:
>
> “In Stevenson et al., it is noted that the clinical annotators all come from the same clinical center [12], this is limitation as changes in national guidelines, clinical training, equipment utilized, and many other factors could influence the data across centers. Future work would benefit from assessing the generalizability of these findings by testing on a held-out dataset recorded at a different center.”
>
> [12]	N. J. Stevenson, K. Tapani, L. Lauronen, and S. Vanhatalo, “A dataset of neonatal EEG recordings with seizure annotations,” Sci. Data, vol. 6, no. 1, p. 190039, 2019.

---

### Decision · Program_Chairs · 2024-09-23

Accept